# Using the Behavioural Regulation in an Exercise Questionnaire (BREQ–2) in Central and Eastern Europe: Evidence of Reliability, Sociocultural Background, and the Effect on Sports Activity

**DOI:** 10.3390/ijerph182211834

**Published:** 2021-11-11

**Authors:** Klára Kovács, Karolina Eszter Kovács

**Affiliations:** 1Institute of Educational Sciences and Cultural Management, Faculty of Arts, University of Debrecen, 4032 Debrecen, Hungary; kovacs.klara@arts.unideb.hu; 2Institute of Psychology, Faculty of Arts, University of Debrecen, 4032 Debrecen, Hungary

**Keywords:** sport motivation, behavioral regulation, higher education

## Abstract

The factors influencing sports motivation create a complex system, integrating internal drivers, such as the love of sport and the need for competence, and external segments, such as the environment, institutions, or the media. In our research, we examined the sports motivation of students studying in higher educational institutions in five countries (Hungary, Romania, Serbia, Slovakia, and Ukraine) using the Behavioral Regulation in Exercise Questionnaire (BREQ–2). This research aims to explore the socioeconomic and institutional factors influencing students’ sports motivation and the effect of sociocultural, demographical, motivational, and societal factors on the frequency of participating in sports. Based on factor analysis, instead of the original five factors, three factors could be detected in the sample: intrinsic and identified motivation, introjected motivation, extrinsic motivation, and amotivation. Based on the results, higher social status facilitates intrinsic motivation, while lower socioeconomic status facilitates extrinsic motivation and amotivation. The strongest effect is exerted by individual variables, of which intrinsic and identified motivation support regular physical activity as a significant factor among students in all countries. This can also be seen in the frequency of sporting activities, as the highest frequency of sports activity was detected among people with high intrinsic motivation.

## 1. Introduction

Although the frequency of sporting activities declines with age, the total drop-out from sport does not occur during the university years, and several students still engage in some form of sporting activity during their university studies. At the same time, drop-out from sport already occurs at a higher rate compared with adolescence, and typically, the interruption of sporting activities is significantly higher after starting work. However, in terms of physical, mental, and social health, maintaining physical activity is desirable for health promotion [1,2]. Therefore, it is essential to examine the motives for participation in sports which may contribute to understanding what underlies the pursuit of sporting activities and for persistence in sports, which may assist in preventing individuals dropping out of sports [3]. In our study, we defined and examined the types of sport motivation with the help of a measuring tool that has not yet been used in the region studied. This research aims to explore the socioeconomic and institutional factors influencing students’ sport motivation and illustrate the peculiarities of sport motivation following the variables mentioned above among students studying in Hungary, Slovakia, Ukraine, Romania, and Serbia. In addition, our aim is to explore the sociocultural and demographical factors and socialization that affect the frequency of students’ sporting activities in each country and the role of sport motivation in addition to these factors.

Generally, motivation can be divided into two large categories, and we can distinguish between intrinsic and extrinsic motivation. However, this two-dimensional grouping does not actually cover the motivational segments with sufficient accuracy, which therefore justifies interpretation in a more detailed category system. For this, one of the most widely used motivational theories is Deci and Ryan’s [4] Self-Determination Theory (SDT), which is based on self-determination. Following this theory, autonomy, competence, and relatedness, as psychological needs, are also detectable segments in sports. Self-determination theory assumes that humans are active beings with an innate tendency for psychological growth and development. The theory’s focus is on the motivation underpinning the behavior, emphasizing the quality rather than the quantity of motivation. The most commonly used grouping follows Deci and Ryan’s organic integration theory (2000), according to which types vary based on the degree and type of the motivational state of the person. Five well-distinguished factors can be identified: lack of motivation (when there is no intrinsic reason to play sports); external regulation (which includes the need to meet some external requirement); introjected regulation (the athletes feel somewhere inside that they need to pursue a sport, but it is not the motivation that becomes internal; rather the external source is replaced by an internal cause, such as the conscience); identified regulation (when the activity is already the result of an individual decision but is still due to the achievement of some goal); and classic intrinsic motivation (when the athletes pursue sports for their pleasure and satisfaction) [4]. Empirical research results show that intrinsic and identified motivation are associated with increased well-being and higher commitment and persistence in sports among adolescents and young adults, while more controlling motives are associated with dropping out of sports over time [5,6,7,8].

Although the study of the social aspect of motivation is not new in research focusing on motivation [9,10,11], it receives less empirical attention than other aspects of motivation, such as competence or performance motivation [12]. The desire for relationships with significant others is central in relation to the social context, as social relationships are fundamentally necessary for optimal psychical functioning, and the need for these relationships provides the energy for social interactions [13]. The social potential of sport provides an opportunity for individuals to create social relationships and experience a sense of belonging. Furthermore, motives for participation in sport and sources of affection show that these social opportunities are significant in terms of participants’ sporting experiences and motivation [14].

Significant others are not limited to peer relationships, as the role of the family as a primary socialisation arena cannot be ignored. The role of parents as extrinsic motivational factors is prominent and is further strengthened by the higher social status of the family. In the case of graduate parents, as well as children from better-off families, this type of motivation is more typical [15,16]. Social and environmental factors can be interpreted as a motivational climate [17], which is significant in several sports contexts (e.g., teammates and sport structures). In addition to the parents, the coach is one of the most important cornerstones of the motivational atmosphere which can increase commitment and persistence in sports [18], following a similar trend in the athletes’ educational institution and the behavior of teachers [19,20]. The aim of our research was to investigate the social differences in the regulation of physical activity and the role of sociocultural, demographic, socialization, and behavioral factors in the prevalence of physical activity across countries. In our research, the BREQ-2 questionnaire was translated and validated into three languages and surveyed in five countries (this measurement tool has not been used in these countries thus far). Then, comparisons were carried out. We looked at its impact on sporting activity, but we also looked at the factors related to social and societal impacts, examining which factors had a significant effect and also their directions. In this way, we also filtered out the effect of other possible factors in addition to motivation.

## 2. Materials and Methods

### 2.1. Sample and Procedure

The PERSIST 2019 (The role of social and institutional factors in student dropout) research was carried out in 2018–2019 among students studying in higher educational institutions of the Northern Great Plain and four cross-border regions (Highlands in Slovakia, Transcarpathia in Ukraine, Vojvodina in Serbia, and Transylvania and Partium in Romania), where Hungarian is the language of instruction. (The former territories of Hungary that now belong to Romania are called Partium and Transylvania, those in the Ukraine are referred to as Subcarpathia, and those in Serbia are called Vojvodina. The students participating in our survey came from these territories, so the names of the countries will be used synonymously with the territories listed earlier. However, it is to be noted that our results and findings only apply to the institutions of these territories, and they are not representative of the entire countries. The following institutions were involved in the study: the University of Debrecen (Debreceni Egyetem) (n = 803), Debrecen Reformed Theological University (Debreceni Református Hittudományi Egyetem) (n = 19), University of Nyíregyháza (Nyíregyházi Egyetem) (n = 112) (Hungary n = 934); Babes-Bolyai University and its outsourced faculties (Babes–Bolyai Tudományegyetem) (n = 163), Emanuel University of Oradea (nagyváradi Emanuel Egyetem) (n = 112), University of Oradea (Nagyváradi Állami Egyetem) (n = 173), Partium Christian University (Partiumi Keresztény Egyetem) (n = 64), Sapientia Hungarian University of Transylvania (Sapientia Erdélyi Magyar Tudományegyetem) (n = 135) (Romania n = 647); Ferenc Rakoczi II Trascarpatian Hungarian College of Higher Education (II. Rákóczi Ferenc Kárpátaljai Magyar Főiskola) (n = 105), Mukachevo State University (Munkácsi Állami Egyetem) (n = 10), Uzhhorod National University (Ungvári Nemzeti Egyetem) (n = 74) (Ukraine n = 189); Constantine the Philosopher University in Nitra (Konstantin Filozófus Egyetem Nyitra) (n = 31), Janos Selye University, Komárno (Selye János Egyetem, Révkomárom) (n = 98) (Slovakia n = 129); and University of Novi Sad, Novi Sad, and Subotica (Újvidéki Egyetem, Újvidék és Szabadka) (Serbia n = 93)) It is one of the easternmost higher educational regions in the European Higher Education Area. The target population was second-year BA and BSc students and second- or third-year students studying a combined (i.e., BA and MA) course. Paper-based questionnaires were completed under the supervision of students participating in courses specifically designed for this research under the instructor’s supervision. Thus, these students served as research staff. For each student participating in a course, we assigned a fellow student from a specific field of study with whom they would complete the specified number of questionnaires.

In the present study, we examined the institutions of the Eastern Hungarian region, as this region has the highest proportion of disadvantaged and non-traditional students at risk of dropping out [21,22]. The Hungarian sample uses quota sampling and is representative of the faculties, the academic area of the courses, and the forms of funding. In the cross-border institutions, we used probabilistic sampling, and students were approached in groups at university and college courses, where they were interviewed comprehensively in the same years. In the case of small Hungarian-language and large mixed institutions, we tried to conduct a complete survey among Hungarian students, while in large Hungarian-language institutions, we tried to collect questionnaires from all fields of study. The total number of items in the sample was 2005.

### 2.2. Instruments

In our research, we surveyed students’ sporting habits with several questions. The most important of these was the frequency of exercise, which we examined with the following question: In the past few months, how often did you perform an intense sporting activity that lasted at least 45 min in addition to physical education courses? The answer options were as follows: never; 1–2 times a year; 1–2 times a month; several times a month; 1–2 times a week; and 3 or more times a week. The answers were recoded as 0–100 points, where 0 meant never, 100 meant three 3 or more, and there were 3 categories of performing a sport: almost never, occasionally, and regularly. The definition of intense sporting activity was based on the hermeneutical approach that the validity of responses is determined by the individual interpretation of the subjects, so we accepted what the respondents thought about as intense sporting activity or the frequency with which it was pursued, when they answered. We examined the level and organizational form of sports with the question of whether he or she was a member of a sports association or club and also indicated the highest level among them, if present, and whether he or she received payment for it. We asked what kind of sport they play most often. Then, we created sports types from the answers, and we examined team and individual sports performers separately.

As an individual level explanatory factor, the Behavioral Regulation in Exercise Questionnaire-2 (BREQ-2, [23]) was applied (Appendix A, Table A1). This measured the levels of self-determination in exercise motivations on a 5-point Likert-scale (from 1 to 5). The questionnaire examines the motivations for exercising in connection with Deci and Ryan’s self-determination theory, and the improved version used in the present study contained 19 items and 5 dimensions: extrinsic (Cronbach α = 0.79), intrinsic (Cronbach α = 0.86), identified (Cronbach α = 0.73), introjected (Cronbach α = 0.80), and amotivation regulation (Cronbach α = 0.83). Factor analysis was applied to divide the question block (maximum likelihood, direct oblimin rotation, KMO = 0.931, explained variance value 58.991%), and 3 factors were obtained (2 original factors were combined), whose factor weights were converted to a scale of 0–100: intrinsic and identified (M = 60.42), amotivation and extrinsic (M = 31.48), and projected regulation (M = 58.77) (Appendix A, Table A2).

Among the social factors, the effect of cultural (parents’ education), economic capital (objective, relative and subjective financial situation), and other socio-demographic background variables (e.g., gender, type of settlement at the age of 14, and parents’ labor market status) on sporting habits were analyzed. Among the sports socialization agents, we explored sports with parents and friends. The effect of the above-mentioned factors on the frequency of sport was examined in a regression model per country in order to explore the role of the types of sport motivation.

### 2.3. Analysis

For the motivational background of sports, we used the Behavior Regulation in Physical Education Questionnaire (BREQ-2). The questionnaire examines the motivation to take part in sports based on Ryan and Deci’s self-determination theory. Previous studies have considered the 19-item measuring tool to be valid and reliable. Originally, it had 5 dimensions, but we determined 3 dimensions for use in our study. These were as follows: intrinsic and identified (Cronbach α = 0.936), amotivation and extrinsic motivation (Cronbach α = 0.881), and introjected regulation (Cronbach α = 0.842). In the course of intrinsic motivation, the individual is driven by internal motivational resources, while the identified regulation can be derived from the recognition and acceptance of the behavior. In extrinsic motivation, the individual wishes to comply with the external environment and circumstances to avoid internal tension or gain social recognition [23,24].

We carried out the analysis of the data with SPSS 24 and applied factor analysis, ANOVA, and linear regression to explore the effect of sociocultural, demographical, individual, and societal factors on the frequency of practicing a sport by country.

## 3. Results

The highest proportion of students (22.4%) pursued sports once or twice a week, 15.7% did so more often, almost the same proportion (19.1%) never did, 10.2% did so annually, 18% pursued them monthly, and 14.6% did so more than once a month. The proportion of students who virtually never practiced any sport was very high, amounting to 29.3%, while 32.6% was the proportion of those who practiced sports occasionally, and 38.1% of the respondents at minimum weekly took part in some sport. Only 3.6% of the students were paid members of some sports club; they were competitive athletes. Meanwhile, 10.6% of the students were members of sports clubs or associations and were not paid for their membership, and 85.9% of them did not belong to any sports organization. Most students performed running and jogging (24.5%), followed by the now classic sport of football at 11.2%, training in a gym (8.6%), and riding a bicycle (7.8%). Smaller was the share of swimming (5.7%) and other sports that are popular nowadays, such as fitness, TRX, and crossfit (4.6%). When sports were arranged into two categories, it turned out that a much higher number of students practiced individual sports (78.3%) than team sports and spectator sports such as basketball, volleyball, and football (22.7%).

When examining social background variables concerning the different types of motivational regulation, most variables played a role in intrinsic and identified motivational regulation. This regulation was most characteristic of men, students of Hungarian nationality, children of parents with a higher educational level and employment status, those living in large cities, and those objectively or relatively well-off financially. In contrast, amotivation and extrinsic motivation were much more common among men and students with lower social statuses, such as children of mothers with primary education, unemployed fathers, those living in villages, those with the most disadvantaged financial situation, and Ukrainian students with Ukrainian nationality, which may be related to the fact that they have the highest proportion of people living in a more modest financial situation. Introjected regulation was more typical of women, Romanian students living in Romania, and children of mothers with secondary education or those whose financial situations were approximately average (Appendix A, Table A3, Table A4 and Table A5).

Significant differences could be detected between the groups formed according to the frequency of sporting activities in the regulation of exercise behavior. Intrinsic and identified regulation were most characteristic of regular athletes (75.9 points), while amotivation and external motivation together and introjected regulation were most characteristic of non-athletes at 34.71 points (F (2.1939) = 22.825, *p* = 0.000) and 66.68 points (F (2.1941) = 146.423, *p* = 0.000), respectively, and intrinsic regulation was least characteristic of them (42.65 points). These results were also supported by Spearman’s rank correlation analyses between the continuous variable of the sporting frequency and the types of control. Sports were moderately positively associated with intrinsic and identified regulation (r = 0.611 **), and a weak negative correlation could be found concerning amotivation and external regulation (r = −0.205 **) as well as introjected regulation (r = −0.368 **).

The representatives of team sports preferred to pursue their chosen type of sport driven by intrinsic motivation (70.4 points) more than those who pursued individual sports (67.1 points), as did unpaid sports club members (77.8) compared with sports club members who also received financial support (73.25) and non-members (59.64 points). The students who were not members of a sports club achieved the highest value in introjected regulation (59.33 points), followed by members without financial benefits (48.87 points) and paid athletes (43.91 points) (F (2.1907) = 53.330, *p* = 0.000). The results illustrate the fact that exercise driven by intrinsic motivation contributed the most to regular sporting activities, mostly in sport clubs. At the same time, we must also realize from the results above that this was not independent of the social background either, as those with a higher social status also had an advantage in this area, which had a direct and indirect effect through motivation as well. The results of intrinsic and identified regulation are summarized in Table 1.

In the final part of our study, we provide an answer to the second research question, (i.e., which explanatory factors have an impact on the frequency of students’ sporting activities in the Hungarian and cross-border areas). To answer this question, we examined the effect of demographic, sociocultural, societal impact, and individual factors on the frequency of sporting activities by country using linear regression with the enter method. Pearson’s correlation showed that all predictor variables correlated weakly or moderately with the independent variables except the type of residential settlement and sporting activity with parents. In all countries, the strongest significant effect was provided by exercise-related regulation (i.e., the individual-level variables, of which intrinsic, identified regulation was outstanding). In all five countries, it had the strongest positive effect, but it was particularly significant for Serbian and Slovakian students (β_Serbia_ = 0.623 (0.536, 1.247) and β_Slovakia_ = 0.499 (0.425, 1.188)), and it is important to note that only did this have an influential role among Slovak students, but the other two types of regulations played a significant role only in the case of Hungarian students, reducing the frequency of sports, which means that a student felt less motivated (β_A&E_ = −0.155 (−0.556, −0.240), β_I_ = −0.216 (−0.528, −0.271)). In addition, the role of the sporting friend as a socialization agent was significant, increasing the frequency of sporting activities among students studying in Hungary (β = 0.132 (5.409, 12.818)), Serbia (β = 0.195 (2.457, 28.861)), Ukraine (β = 0.193 (2.708, 24.564)), and Romania (β = 0.074 (0.363, 10.006)). Among the students studying in Romania and Ukraine, the mother’s employment status had a different effect; it decreased the frequency of taking part in sports among the Romanian students (β = −0.80 (−10.892, −0.120)) and increased it among the Ukrainian students (β = 0.175 (0.665, 24.170)). 

In the Romanian and Serbian subsample, the impact of gender was significant too. In both cases, being female reduced the frequency of physical activity (β_Romania_ = −0.11 (−14.205, −3.612), β_Serbia_ = −0.180, (−37.399, −2.503)) (Appendix A, Table A6).

## 4. Discussion

Our research examined the sports motivation of higher educational students and the factors influencing it, based on the Behavioral Regulation in Exercise Questionnaire-2 (BREQ-2) of Marklan and Tobin [1]. The adaptation of the questionnaire had not been carried out in the studied region until the present research. However, based on the reliability studies, the questionnaire can be reliably applied. The Hungarian version of the questionnaire could be divided into three factors: the internal and identified subscales, the amotivation and external motivational subscales, and the introjected regulation subscales.

Based on the results, a higher level of intrinsic and identified regulation was detected among men, students with Hungarian nationality and learning in Hungary, children of parents with a higher educational level and employment status, those living in bigger cities, and those characterised by an objectively or relatively well-off financial situation. Thus, it seems that a better financial background mostly allowed and facilitated intrinsic motivation, which can be of paramount importance for the effectiveness of sports activities. In contrast, amotivation and external regulation were typically associated with lower a socioeconomic status. A lack of sports motivation may coincide with a lower sporting frequency, which is often due to insufficient financial support. However, it is worth comparing the trend during higher educational years with high school practice. In the case of high school students, parent-facilitated sports participation, which can be considered a strong external motivational segment, was typically stronger among those with better financial statuses. Presumably, however, early external motivation becomes internal over time, so those who initially played sports only because of their parents’ encouragement will continue to experience it as adults as a source of their own pleasure [25]. From the socio-demographic point of view, introjected regulation is most characteristic of women and students with a relatively average financial status. Thus, socioeconomic status variables are determinant in sports motivation, and a low status hinders the completion and internalization of sports motivation. Due to financial problems, some students are unable to mobilize resources for a higher level of commitment, so their interest in sports will be lost. Presumably, a smaller group may be driven by financial reasons (e.g., sports club membership with financial benefits), which may, however, reduce intrinsic sports motivation in some cases by giving way to extrinsic sport motivation [26]. In this capacity, intrinsic motivation is mostly associated with Riesman’s [27] theory of inner-directedness or Rotter’s [28] internal locus of control, where the main source is the person’s intrinsic driving force, self-compliance, responsible behavior (in this case, being health-conscious), and a high level of perceived self-control. This is also reflected in the frequency of sporting activities, as the highest frequency of sport was detected among people with intrinsic motivation, which can stem from their personalities and attitudes toward sports while supported by a better financial background. Its significant and moderately strong relationship with intrinsic and identified regulation also strengthens this picture, while its weak negative correlation with amotivation and external regulation as well as the introjected motivational factor suggests that sporting activity takes place in a less determined way with a lower frequency [8,29].

Overall, we can see that social status, and the cultural capital and economic situation within it, are significantly associated with exercise-related regulation. Children of parents with a higher status (with the associated higher cultural capital) are more able to recognize, use, and ultimately internalize their beliefs and the importance of sport, while financial difficulties associated with lower social status contribute to losing interest in sports or exercising due to compliance [25,30]. A movement for social recognition associated with exercise or the reduction of internal tension was more of a middle class feature among students. Those who are characterized by intrinsic, identified regulation show similar characteristics to Riesman’s inner-directedness, and those with extrinsic and introjected regulation show similarities with the other directed type [27]. An inner-directed person has a solid personality and values, while the other controlled elements of people’s behaviors and lifestyles, and thus their relationships to sports, are determined by the external environment and compliance with others. In relation to sports, inner-controlled people pursue sports through internal motivation, while other controlled people want to meet social and micro-environmental expectations (e.g., health beliefs) or the image and fashion suggested by the media. Of the internal motivations, sporting activities contribute the most to regular and more club-based sporting activities, but the role of social background should be emphasized, which provides a huge benefit for students who can be characterized by a better socio-economic status [31].

In addition, in our research, we examined the influencing role of demographic, sociocultural, societal, and individual factors on the frequency of sport by country. The strongest effect was given by the individual variables, of which the internal, identified motivation was the strongest, which supports regular physical activity as a significant factor among students in all countries. External regulation and amotivation, as well as introjected regulation, on the other hand, were rather hindering factors, but their effect was significant only in the Hungarian subsample. These factors interact in a circular way, as a higher frequency of sporting activities further strengthens intrinsic and identified motivation and thus sports persistence [29,32] and, at the same time, reduces the level of introjected and external regulation and amotivation [33].

The role of social influences, including peer influence, should be emphasized. In all countries except for Slovakia, it had a very important positive effect on students’ sporting activities. In adolescence and after, the social network is restructured. Instead of parents, peers become more and more important, and as a reference group, peer communities will have a primary role in socialization, including sports socialization. The most important factor is the joy and experiences gained during joint sporting activities, as the lack of these can encourage people to quit. In this way, peers and sports counterparts also shape sporting habits [34]. Thus, the role of social support also appears in relation to peers, and although not with such a specific weight, it points to the protective and retaining power of a supportive environment [15,35].

We obtained the opposite result regarding the labor market status of the mothers, which designates further research directions. However, it is likely not the employee’s status itself that plays a role, but the type of work, the position in the labor market, and the income associated with it. A higher income and financial well-being are likely to contribute to children’s sporting opportunities in these families, while families receiving lower incomes have fewer opportunities for certain forms of sport or any sport at all, which in most cases involve additional costs (e.g., membership fees, equipment, travel, and camps).

Numerous studies have shown gender differences in sports (significantly fewer women play sports than men) and draw attention to the need for prevention programs to involve more women in the world of sports [16]. Our results also confirmed the disadvantage of Romanian and Serbian women in the frequency of sports, in which the fact that they were more characterized by the introduced regulations also played an important role. This result is consistent with research findings that call attention to motivational differences by gender in sport. While girls tend to play sports to be healthy, pretty, look good, and meet the requirements of health awareness, which act as external motivating factors, for boys, victory, competing, and demonstrating their strength are important motivating forces, but maintaining health as a motivation is not negligible among them either [29,36,37].

## 5. Conclusions

Our research explored the social differences in the regulation of physical activity and the role of sociocultural, demographic, socialization, and behavioral factors in the prevalence of physical activity across countries. Our main findings show that intrinsic, identified regulation is more likely to be found in people from higher social backgrounds and men, and introjected regulation is more typical of middle class people, while extrinsic regulation and amotivation is more common among women and people from lower social backgrounds. Intrinsic, identified regulation had the largest effect on the frequency of sports participation in all countries when controlling for the effect of all explanatory variables, while the other two types of regulation had a negative effect. Thus, the more likely one pursues a sport due to external influences, possibly such as social reinforcement, or to avoid remorse, the less often they will do it, and the higher the likelihood of dropping out of it becomes. This is also important because these factors have an influential effect in filtering out the impact of social and societal backgrounds. Moreover, in some cases, institutional-level variables related to sports are able to override the disadvantages of a low socio-economic status [31]. Our results also show that having a sporting friend has one of the most influential roles on the frequency of sporting activities. Therefore, regular exercise is best achieved by students with a friend with whom they can play sports who, if necessary, motivates the student and gives strength and a pattern to the sport.

## Figures and Tables

**Table 1 ijerph-18-11834-t001:** Means of intrinsic and identified regulation according to sporting habits (on a scale of 0–100) Source: PERSIST 2019.

		Intrinsic and Identified Regulation	F	sig	N
**Frequency of sport**	almost never	42.65	472.244	0.000	1941
	rarely	60.58			
	regularly	75.9			
**Sport club membership**	member without financial benefit	77.8	69.895	0.000	1909
	member with financial benefit	73.25			
	non member	59.64			
**Type of sport**	individual	67.1	6.152	0.013	1291
	team	70.4			

## Data Availability

Data are available only on request due to ethical restrictions.

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
