# Peer review of "Using the Behavioural Regulation in an Exercise Questionnaire (BREQ–2) in Central and Eastern Europe: Evidence of Reliability, Sociocultural Background, and the Effect on Sports Activity"

_ijerph, 2021, doi:10.3390/ijerph182211834_

Round 1
Reviewer 1 Report
Overall Comments:
This study evaluates different types of behavioral regulation of exercise behavior, the relationships of the identified behavioral regulation types (intrinsic/identified, projected, amotivation/extrinsic) with socio-demographics and exercise frequency, in a population of Central and Eastern European university age students. The manuscript would benefit from some reorganization and additional methodological details.
Major remarks:
- The introduction is very long, and a bit difficult to follow (the ordering of the contents does not seem optimal). The literature review and theoretical model can build to the study aims. At a minimum, the description of the study aims (lines 35-43) should occur toward the end of the introduction.
- Line 143: sporting activity needs to be defined? Were the respondents given any detail as to what counts as “intense sporting activity”?
- Line 144: 45-minute bouts of intense activity are very long. I don’t see any presentation of the frequencies for this variable (which is one of the study outcomes) presented anywhere.
- Results: Data in the tables needs to be presented in the same order as the data in the text (for example, Table 1 data should be the first data described in the results). Tables should be renumbered accordingly.
- Methods, Analysis: Much more detail is needed for each regression model. What was the dependent variable? What were the independent parameters (exposure of interest and covariates that were adjusted for)? What type of regression model was used?
Minor remarks:
- Lines 20-22: Seems like it might work better if moved to before sentence starting on line 19.
- Lines 125-129: This description confused me. The study used students from specific courses to collect data on other university students?
- The citation style (in text) doesn’t match the reference list style.
- Line 257 (and throughout)- the authors discuss “retention of intrinsic motivation”, but it is unclear how retention is being assessed. The data are cross-sectional, so I am not sure retention (no decline over time) can be measured with these data.
- Line 292 (and elsewhere)- the word “influences” implies the direction of an association identified (for example, socioeconomic status influences behavioral regulation), but as these data are cross sectional, the direction cannot be ascertained (you do not know for sure that behavioral regulation does not influence socioeconomic status). May want to consider replacing this language with “is associated with”
Author Response
Dear Reviewer!
Thank you very much for your evaluation! We hereby summarise our answer and modifications:
A. Major remarks:
- We shortened the Introduction part and highlighted the aims of the study at the end of the section.
- We defined the term 'intense sporting activity' (Line 157-160)
- We presented the frequencies of intense sporting activity at the beginning of the Results section
- We renumbered the tables
- We added the requested details concerning the regression model
Minor remarks:
- Lines 20-22: we changed the order of the mentioned sentences
- Lines 125-129: we clarified the role of the mentioned students who served as a research staff
- We revised the citation cite and used the one requested by the journal (numbered citation)
- we revised the question of 'retention of intrinsic motivation'
- the word 'influences' is replaced with 'is associated with'
Reviewer 2 Report
This is an interesting paper with valuable insight in to a much needed area of research. The construction of the paper needs attention on one point - that is, there needs to be a clear conclusion to this work. The discussion section merely transfers to the references. I would like to see some summative commentary about the impacts of your research in a conclusion section. There is some commentry in the discussion section which might assiste here. The abstract also would benefit from a stronger indication of what the conclusion of the authors are from this extensive research.
Author Response
Dear Reviewer!
Thank you very much for your evaluation! We hereby summarise our answer and modifications:
- we highlighted some summative commentary about the impacts of your research in a conclusion section
- we also highlighted this in the abstract
Thank you very much for your evaluation!
Best wishes,
the Authors